# Reliability Evaluation for Cable-Spring Folding Wing Considering Synchronization of Deployable Mechanism

**Yun Gao [1], Ming Hu [1,\*], Xiaohong Zhou [2] and Mingzhong Zhang [2]**

1   School of Mechanical Engineering & Automation, Zhejiang Sci-Tech University, Hangzhou 310018, China; 201810501016@mails.zstu.edu.cn
2   Beijing Research Institute of Mechanical & Electrical Engineering, Beijing 100074, China; hebeisongwei@163.com (X.Z.); zhmzhbj@163.com (M.Z.)
\*   Correspondence: huming@zstu.edu.cn

**Abstract:** The cable-spring folding wing is a novel type of rigid-flexible coupling structure for missiles, which is composed of several sets of deployable mechanisms, with each composed of a wheel-rope transmission system and a parallel spring driving mechanism. The movement of the cable is initiated by the driving force produced by parallel springs, which directly changes the magnitude and the distribution of the driving force. Therefore, the cable-spring folding wing system has the typical characteristics of strong nonlinearity and motion coupling. In addition, each deployable mechanism shares an identical structure, but the distribution of motion parameters is discrepant due to external loads. Asynchronous movement of the cable-spring folding wing will occur and become a significant issue, which is detrimental to the working performance and could even lead to failure. Focusing on these problems, the multi-body dynamics theoretical model and simulation model of deployable mechanism are established, the kinematic and dynamic characteristics of critical components are studied, and the key factors affecting the deployment performance are investigated. A new reliability method with an angular precision control for deployable mechanism is proposed based on the theoretical model. The effectiveness of the proposed model and method is verified by comparing it with the Monte Carlo method. On this basis, the reliability evaluation for cable-spring folding wing, comprehensively considering deployment performance and synchronization, is carried out.

**Keywords:** cable-spring folding wing; rigid-flexible coupling; deployable mechanism; synchronization; precision control; reliability method





## 1. Introduction

Link, torsional spring and cable folding wings are frequently utilized to improve a missile's loading capacity and enhance its combat capability [1]. The folding wing resting in the launch tube is initially constrained by the inner wall at a certain angle. Then, the unconstrained folding wing deploys rapidly under the action of the deployable mechanism during ejection of the missile [2]. Subsequently, the locking pins are engaged to lock up the folding wing in a timely and accurate manner. Therefore, the operational capability of folding wings is strongly dependent on the performance of deployable and locking mechanisms [3].

Scholars have undertaken many studies on folding wings, including structural design and analysis, dynamic modeling, simulation and tests, aeroelasticity and flight stability control, and reliability calculation and evaluation [4–10].

Kroyer designed a novel torsional spring-cable folding wing and analyzed its structural characteristics, such as strength, stiffness and flutter, using ADINA. The results show that the designed deployable mechanism could satisfy the requirements [11]. Based on the aerodynamic instability and beam theory, Wang et al. established an aeroelastic model of a torsional spring folding wing, and a theoretical method was proposed to predict the natural

and flutter frequency in different segments at various angles under diverse aerodynamic conditions [12,13]. With the help of finite element software, Coffin et al. analyzed the dynamic characteristics of the link folding wing. It was found that the results obtained from the simulation had a high consistency with the experimental values [14].

In the studies of the folding wing of an anti-tank missile, Harris et al. established the attacking flight dynamics model and simulated several fault modes by the Monte Carlo method; he subsequently found the weakness of the folding wing [15]. Liao et al. developed a simplified model to evaluate the fatigue life of folding tail with multiple cracks. The stress state of the folding tail with multiple cracks under a different load spectrum was obtained by the finite element method, and the total fatigue life of the folding tail was predicted effectively [16]. Hu et al. introduced a parameterized simulation model of deployable mechanism, taking errors in the dimension, clearance and the peak of driving force into account. By using the ADAMS/Insight, a reliability simulation test was carried out to analyze and evaluate the reliability of the link folding wing. Unfortunately, this method was time-consuming, and thus unsuitable for strongly nonlinear models [17]. Xie et al. proposed a new reliability model of a torsional spring folding wing, considering the common cause failure effect, and the availability of this method was examined using the Monte Carlo simulation method [18]. Nevertheless, the reliability index of folding wing only concerns the deployment time, ignoring the asynchronous problem caused by the deployment time difference. Therefore, in view of the synchronization reliability problem, Yu et al. proposed a novel synchronization reliability evaluation method by dividing the integral domain into several independent domains, which can more accurately evaluate the reliability of folding wings [19]. However, the structural details and loading condition of deployable mechanism were neglected; instead, an agent model was used in this approach.

In the relevant literature, the driving devices commonly applied in the deployable mechanisms are motors, torsional springs, gas or hydraulic actuators. For stability and manipulation purposes, the transmission devices are mostly links, rods or crank sliders. Their dynamic models are relatively simple because of their rigidity, but prone to being locked mechanically due to their complicated structure. In addition, there is no dead point or interference in rigid-flexible coupling folding wings. Moreover, there is little publicly available literature on rigid-flexible coupling deployable mechanisms of folding wing, and these are not only old but also do not offer a detailed analysis. The cable-spring deployable mechanism of folding wing is a novel type of rigid-flexible coupling structure, using a wheel-rope transmission system and a parallel spring driving mechanism [19]. The application is related to national defense and security, and has the characteristics of simple structure, flexible controllability and high reliability. Therefore, it is necessary to study the dynamic characteristics of the cable-spring deployable mechanism.

In addition, the reliability indexes of the folding wing introduced in the literature mainly include the motion parameters of the deployment stage, such as deployment time or deployment angle, and the requirements of the locking stage, such as positioning accuracy, locking reliability and vibration intensity. The structures of four sets of cable-spring deployable mechanisms of the 'X' type folding wing studied in this paper are identical, but the deployment times of each set are different due to the diversity of loads. If the deployment time difference in the prescribed position is too marvelous, it is possible that the missile may fail. Therefore, it is vital to comprehensively consider deployment performance and synchronization for reliability.

On the basis of rigid-flexible coupling dynamics analysis of the deployable mechanism, a novel reliability evaluation method for the cable-spring folding wing is proposed, which has applicable values for solving the key problems, such as its being time-consuming and incomplete evaluation. The literature [2] has discussed the working principle of the folding wing, analyzed the dynamic characteristics while neglecting the friction torque, and calculated the reliability considering only three factors. Based on the previous related research and focusing on the phenomenon of asynchrony between the upper and lower wings, this paper provides a novel idea for comprehensively evaluating the reliability of

cable-spring folding wing. The present research includes three parts: the research status of folding wing, rigid-flexible coupling theoretical modeling and dynamic simulation of the deployable mechanism and comparison of the results, and a reliability evaluation for the cable-spring folding wing with precision control.

## 2. Dynamic Theory of Deployable Mechanism

### 2.1. Working Principle

The structure of the research object in the present research is same as that in [2]. Each cable-spring folding wing is mainly composed of a group of lateral deployable mechanism and two groups of longitudinal locking mechanisms. Additionally, the lateral deployable mechanism is composed of a wheel–rope transmission system and a parallel spring driving mechanism, as shown in Figure 1.

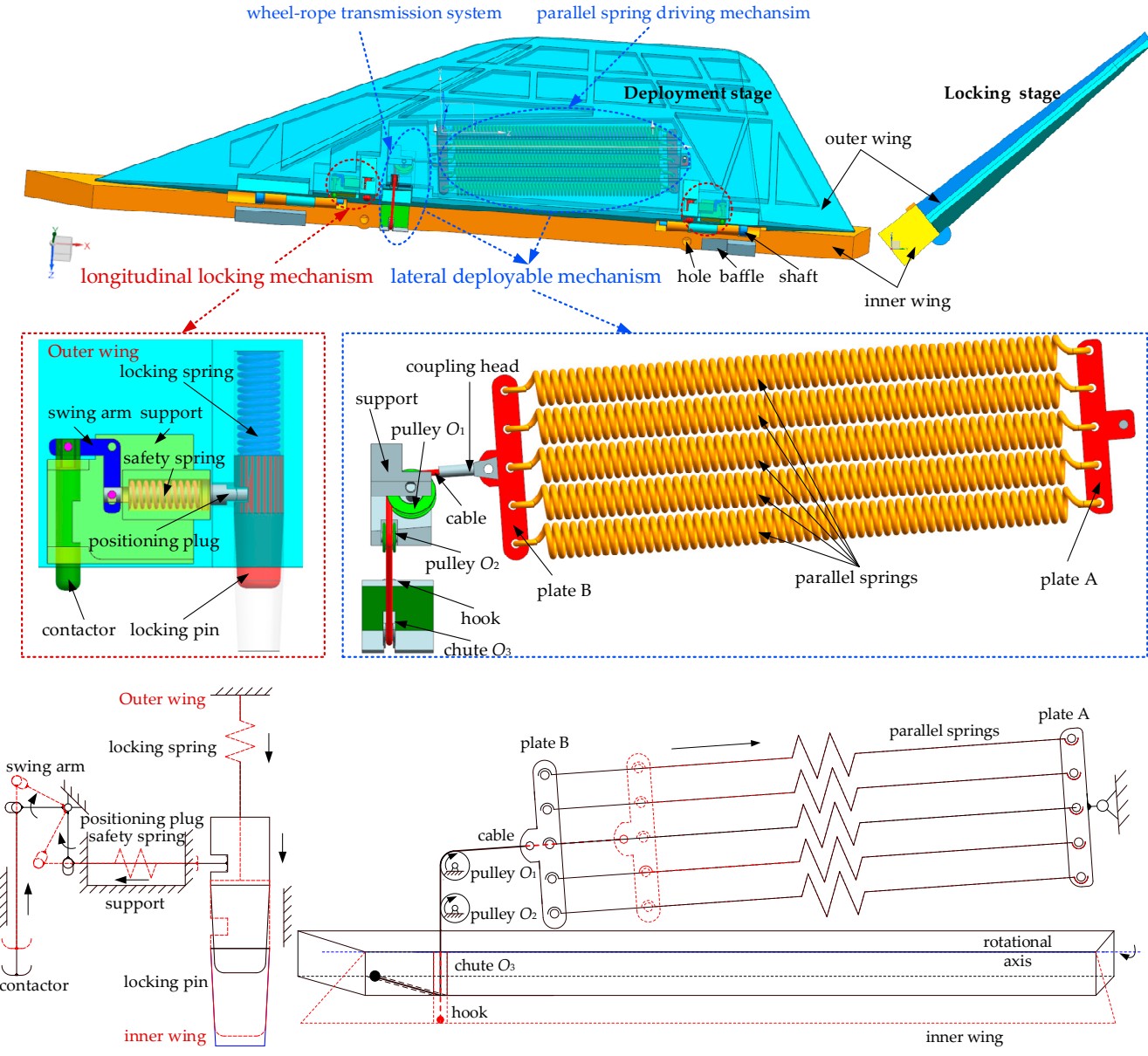

**Figure 1.** Structure of folding wing.

The folding wing is released when the missile is launched, then the deployable mechanism is initiated. The driving torque produced by the wheel-rope transmission system and parallel springs driving mechanism drives the folding wing to deploy rapidly. Before

the folding wing deploys to the designated specified position, the contactor collides with inner wing, which triggers the locking mechanism. Since the positioning plug withdraws from the groove of the locking pin, the released locking pin enters the locking hole rapidly. Thus, the deployment and locking stages of the folding wing are completed.

### 2.2. Theoretical Modeling

As shown in Figure 2, the folding wing is subjected to the driving torque $M_d$ produced by the parallel compressed springs, the gravity torque $M_g$ of the outer wing and its accessories, the friction torque $M_f$ on the rotating pair, the axial and normal aerodynamic torque $M_q$ during deployment.

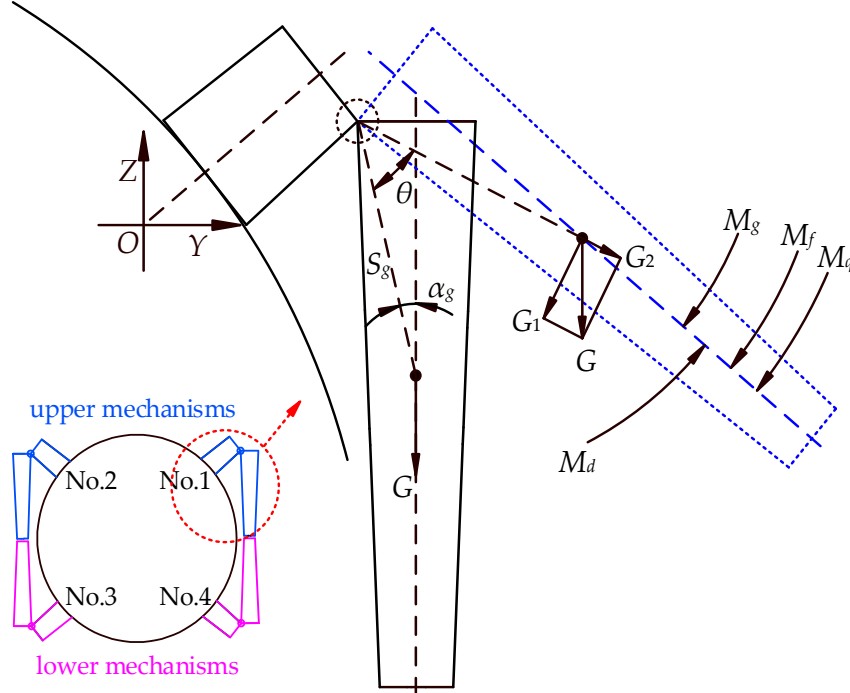

**Figure 2.** Force diagram of folding wing.

In the theoretical model, if the change in the center of gravity $O_g$ of the outer wing and its accessories caused by the deformation of the parallel springs and the influence of the weight of the cable and aerodynamics force during the deployment stage are ignored, taking the deployment angle as a generalized coordinate, the dynamic differential equation of the folding wing can be expressed as

$$J_1\ddot{\theta} = M_d - M_g - M_f, \tag{1}$$

where $\ddot{\theta}$ is the deployment angular acceleration, and $J_1$ is the rotational inertia of the outer wing and its accessories.

It should be pointed out that Formula (1) is available for the upper wings, because the gravity torque retards the deployment of the upper wings (No. 1 and No. 2) while promoting the deployment of the lower wings (No. 3 and No. 4). When establishing the equation for the lower wings, the positive-gravity torque is needed to replace the negative gravity torque on the right side of the equation.

The driving torque $M_d$ can be expressed as

$$M_d = F_s d, \tag{2}$$

where $F_s$ is the driving force and $d$ is the arm.

Assuming that the force state of each parallel spring is exactly same, the driving force $F_s$ can be expressed as

$$F_s = n \cdot (f_0 - k_s \cdot \Delta l), \tag{3}$$

where $n$ is the number of parallel springs; $f_0$ and $k_s$ represent the prestressing force and stiffness coefficient of each parallel spring, respectively; $\Delta l$ is the deformation of spring.

According to the structure of the folding wing and the working principle of the deployment stage, the outer wing is initiated by the retraction of the parallel springs. The retraction drives the cable wound on pulleys ($O_1$ and $O_2$) and chute $O_3$ to move, and the movement of the cable compels the outer wing to rotate around the axis, which directly leads to the change in the direction and the magnitude of the driving force, as well as the change in the magnitude of the arm. That is, the driving force $F_s$ and arm $d$ vary with the deployment angle $\theta$. Therefore, the deployable mechanism transforms the linear motion of the parallel springs into the rotational motion of the folding wing through the wheel-rope transmission system, which implements the transformation of the form of motion and the transmission of the energy of motion.

As shown in Figure 1, the relative position of pulley $O_1$ and pulley $O_2$ is definite, since both are installed on the outer wing. This means that the tangent section of the two pulleys does not change with the deployment angle, and the relationship between the hook and chute $O_3$ mounted on the inner wing is the same as them. In addition, chute $O_3$ is coplanar with the pulley $O_2$, and both are perpendicular to the rotational axis. If the rotational axis is defined as the $x$-axis, the equivalent model of deployable mechanism is shown in Figure 3.

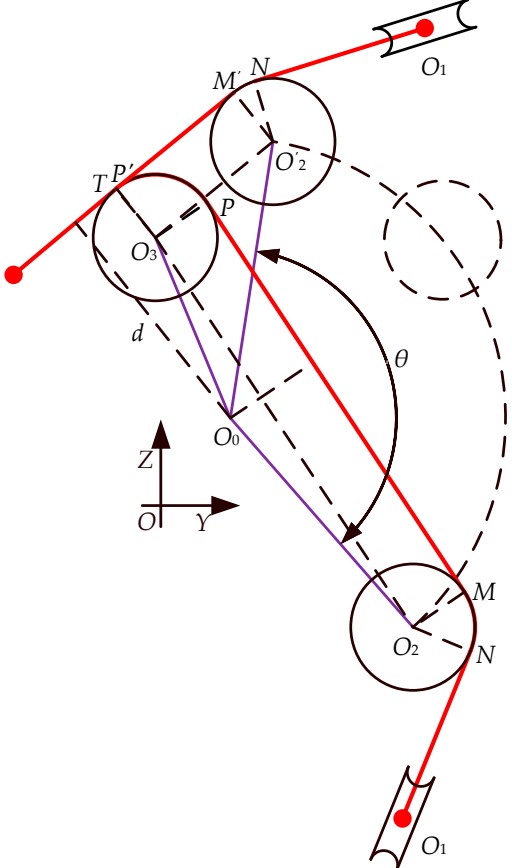

**Figure 3.** Equivalent model of deployable mechanism.

Here, $O_0$ $(y_0, z_0)$, $O_1$ $(y_1, z_1)$, $O_2$ $(y_2, z_2)$ and $O_3$ $(y_3, z_3)$ are projection positions of the cross-sectional center of the shaft, pulley $O_1$, pulley $O_2$, and chute $O_3$ on the *oyz* plane; $R_2$ is the equivalent radius of the pulley $O_2$ and the cable; $R_3$ is the equivalent radius of the

chute $O_3$ and the cable; $N$, $M/M'$ or $P/P'$, $T$ are the tangent points of the pulley $O_1$ and the pulley $O_2$, the pulley $O_2$ and the chute $O_3$, the chute $O_3$ and the hook, respectively.

If the change in the length of the cable caused by the elastic deformation is ignored, the deformation $\Delta l$ can be expressed as

$$\Delta l = \Delta MP + \Delta \widehat{NM} + \Delta \widehat{PT}, \tag{4}$$

where $\Delta l$ is equal to the variation of the winding length between the pulley $O_2$ and the chute $O_3$, and deformations can be expressed as

$$\Delta MP = \Delta O_2 O_3 = O_2 O_3 - O_2' O_3, \quad \Delta \widehat{NM} = \widehat{NM} - \widehat{NM'}, \quad \Delta \widehat{PT} = \widehat{PT} - \widehat{P'T}. \tag{5}$$

$R_2$ is equal to $R_3$ and the chute $O_3$ is coplanar with the pulley $O_2$. Therefore, they can be expressed as

$$
\begin{aligned}
O_2 O_3 &= \sqrt{(y_2 - y_3)^2 + (z_2 - z_3)^2}, \\
O_2' O_3 &= \sqrt{(y_2' - y_3)^2 + (z_2' - z_3)^2},
\end{aligned}
\tag{6}
$$

where

$$
\begin{aligned}
y_2' &= y_0 + \sqrt{(y_2 - y_0)^2 + (z_2 - z_0)^2} \cdot \cos(\theta - \arctan \tfrac{z_2 - z_0}{z_2 - z_0}), \\
z_2' &= y_0 + \sqrt{(y_2 - y_0)^2 + (z_2 - z_0)^2} \cdot \sin(\theta - \arctan \tfrac{z_2 - z_0}{z_2 - z_0}).
\end{aligned}
\tag{7}
$$

Additionally,

$$
\begin{aligned}
\widehat{NM} - \widehat{NM'} &= R_2 \cdot (\angle NO_2 M - \angle NO_2 M'), \\
\widehat{PT} - \widehat{P'T} &= R_2 \cdot (\angle PO_3 T - \angle P'O_3 T).
\end{aligned}
\tag{8}
$$

Here,

$$\angle NO_2 M - \angle NO_2 M' + \angle PO_3 T - \angle P'O_3 T = \theta. \tag{9}$$

Therefore, the change in length of the cable wound on the pulley $O_2$ and the chute $O_3$ can be expressed as

$$\Delta \widehat{NM} + \Delta \widehat{PT} = R_2 \theta. \tag{10}$$

The deformation $\Delta l$ can be obtained by combining the Formulas (4)–(7) and (10).

The arm $d$ can be expressed as

$$d = \frac{O_3 O_0 \cdot O_2' O_0 \cdot \sin(\theta_0 - \theta)}{O_2' O_3} + R_2, \tag{11}$$

where

$$
\begin{aligned}
O_2' O_0 &= \sqrt{(y_2 - y_0)^2 + (z_2 - z_0)^2}, \\
O_3 O_0 &= \sqrt{(y_3 - y_0)^2 + (z_3 - z_0)^2}, \\
\theta_0 &= \tfrac{3\pi}{2} - \arctan \tfrac{z_3 - z_0}{y_0 - y_3} - \arctan \tfrac{y_2 - y_0}{z_0 - z_2}.
\end{aligned}
\tag{12}
$$

This means that the functional relationship between the deployment angle $\theta$ and the driving torque $M_d$ is determined.

When conducting a ground test, the inner wing is commonly fixed on the missile. The outer wing is connected to the inner wing through a rotating pair and is perpendicular to the ground. Therefore, the gravity torque can be expressed as

$$M_g = m_1 g s_g \sin(\theta + a_g), \tag{13}$$

where $m_1$ is the mass of the outer wing and its accessories; $g$ is the acceleration of gravity; $s_g$ is the distance between the center of gravity $O_g$ and $O_0$; $a_g$ is the angle formed by the line $O_g O_0$ and $z$-axis, as shown in Figure 2.

In addition, the friction torque $M_f$ on the rotating pair can be expressed as

$$M_f = r_f f_v R_f, \tag{14}$$

where $r_f$ is the radius of friction circle; $f_v$ is the coefficient of equivalent friction; $R_f$ is the reaction force, which can be expressed as

$$R_f = \sqrt{G^2 + F_s^2}. \tag{15}$$

### 2.3. Solution of Theoretical Model

As shown in Figure 4, the curves between the driving force $F_s$, the arm $d$, the driving torque $M_d$, the gravity torque $M_g$, the friction torque $M_f$, the resultant torque $M$ and the deployment angle $\theta$ are obtained using the above formulas.

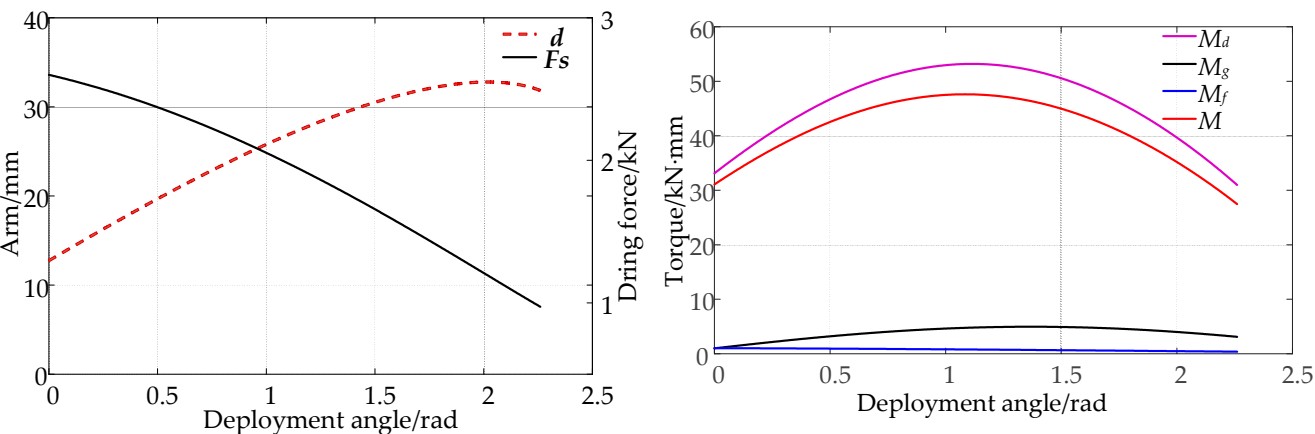

**Figure 4.** The relationship between driving force, arm, torque and deployment angle.

Formula (1) is a second-order nonlinear differential equation of the deployment time $t$. Among them, the prescribed position $\theta_p(t_e)$ is equal to 129.5°, and both the initial deployment angle $\theta(t_0)$ and the initial deployment angular velocity $\omega(t_0)$ are equal to zero. According to the numerical method of Runge—Kuta, the high-precision ODE 45 can be employed to solve the theoretical model of deployable mechanism. As shown in Figure 9, the relationships between deployment angle, deployment angular velocity and deployment time are obtained.

The results shows that the angular velocity of the folding wing increases during deployment, regardless of the influence of gravity torque $M_g$ or friction torque $M_f$. Friction and gravity torque can extend the deployment time, but the deployment angular velocity decreases, which reduces the impact force of collision between the outer wing and inner wing. Furthermore, it is found that the influences of gravity torque on motion parameters are more significant.

## 3. Dynamic Simulation

### 3.1. Simulation Model

The first step is importing and assembling the 3D model of deployable mechanism in the environment of ADAMS/View. A fixed pair is used in the ground and inner wing and a rotating pair is used in the inner wing and outer wing.

The second step is creating a parallel spring driving mechanism. The parallel springs are used as a power source to connect the outer wing and one hand of the cable, and the other hand of the cable is fixed on the inner wing. The cable is wound on two pulleys and a chute. Thus, generation of the parallel springs in terms of parameters such as $f_0$, $k_s$ and installation position is required.

The third step is the generation of a wheel-rope transmission system. One can create related parts by calling the cable module in ADAMS. It is important to set appropriate parameters in terms of diameter, density and the Young's modulus of the cable. Additionally, it is also necessary to set the section properties (such as width, depth, radius, angle), spatial layout (such as position, diameter, deflection angle), connection (such as connection type, parts), and the density of the pulleys and chute according to Table 1. Combining these parameters with the start point, the end point and winding sequence of the cable can generate a wheel-rope transmission system, as shown in Figure 5.

**Table 1.** Sectional properties of pulleys.

| Parameter | $O_1$ | $O_2$ | $O_3$ |
|---|---|---|---|
| With/mm | 7.0 | 7.0 | 6.6 |
| Depth/mm | 3 | 1.8 | 1.8 |
| Radius/mm | 2.0 | 2.0 | 2.0 |
| Angle/° | 20.0 | 20.0 | 20.0 |
| Location | $O_1$ | $O_2$ | $O_3$ |
| Diameter/mm | 25.6 | 16.0 | 16.0 |
| Mis-alignment X/° | 66.6715 | 0.0 | 0.0 |
| Mis-alignment Y/° | 7800 | 7800 | 7800 |
| Joint Type | Fixed | Revolute | Revolute |
| Connection Part | Outer wing | Outer wing | Inner wing |

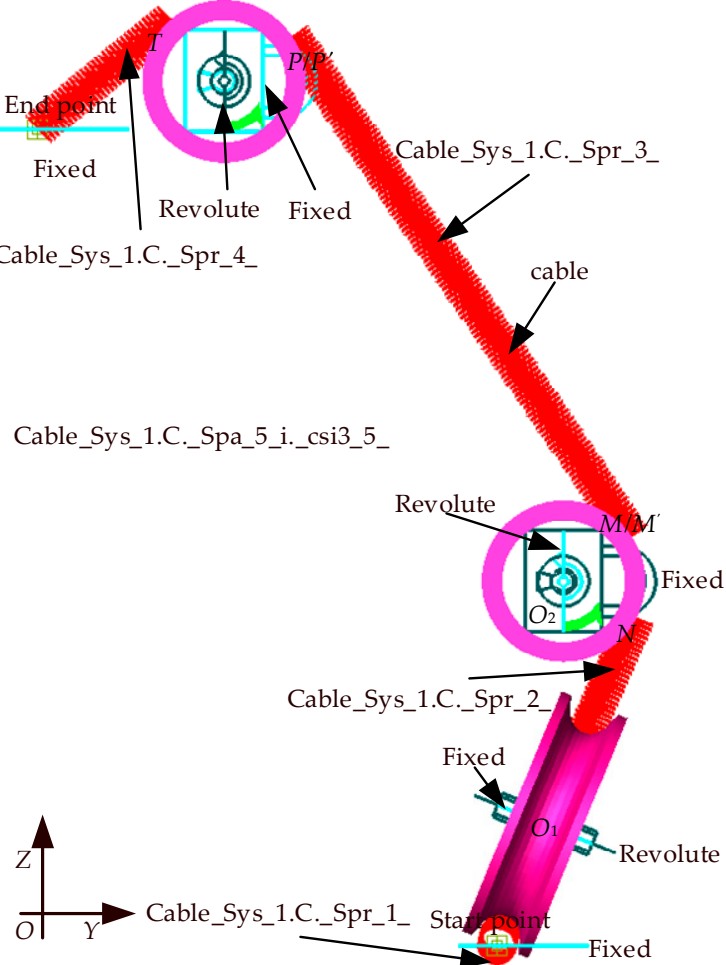

**Figure 5.** Simulation model of transmission system.

The next step is establishing the locking springs and contact relationships, such as the relationship between the contactor and inner wing, positioning plug and locking pin, locking pin and inner wing, and lock pin and outer wing.

The last step is setting the running time of the simulation model to 0.15 s, the number of steps to 15,000, the solver type to WSTIFF/I3, and the integral error to 0.001, according to the deployment time of the theoretical results.

### 3.2. Solution of Dynamic Simulation Model

As shown in Figure 6, the curves represent the relationship between the deformation velocity of each compressed parallel spring, equivalent spring and deployment time, respectively.

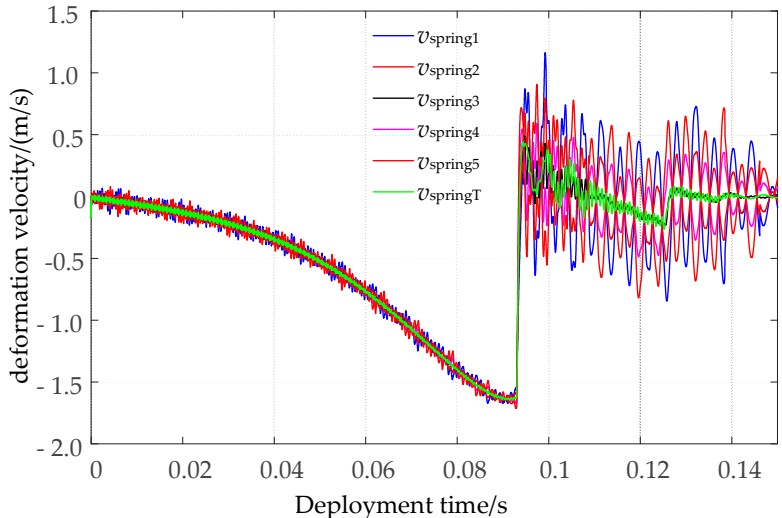

**Figure 6.** The curves in the deformation velocity of springs and deployment time.

The results show that the simulation model of the parallel springs could be replaced by an equivalent spring. The deformation velocity of the springs fluctuates in both the deployment and locking stages. In addition, the fluctuation in the initial phase of the locking stages becomes more significant than in the deployment stage. However, it tends to become gradually stabilized in the deployment stage, and close to zero in the locking stage with deployment.

Figure 7 displays the relationship between the deployment angle, deployment angular velocity and deployment time.

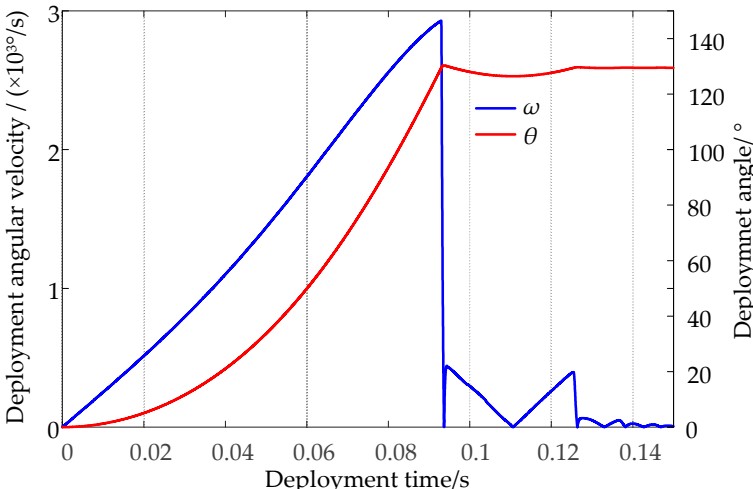

**Figure 7.** The relationship between deployment angle, deployment angular velocity and time.

The results show that when the angle reaches the prescribed position, the time is equal to 0.0921 s, and the outer wing collides with the inner wing. At the same time, the deployment angular velocity reaches the maximum value, namely, 2924°/s; In addition, the outer wing moves backward slightly when the deployment angle reaches 130.2° since the inner and outer wing penetrate into each other.

The relationships between the displacement of locking springs and deployment time are shown in Figure 8.

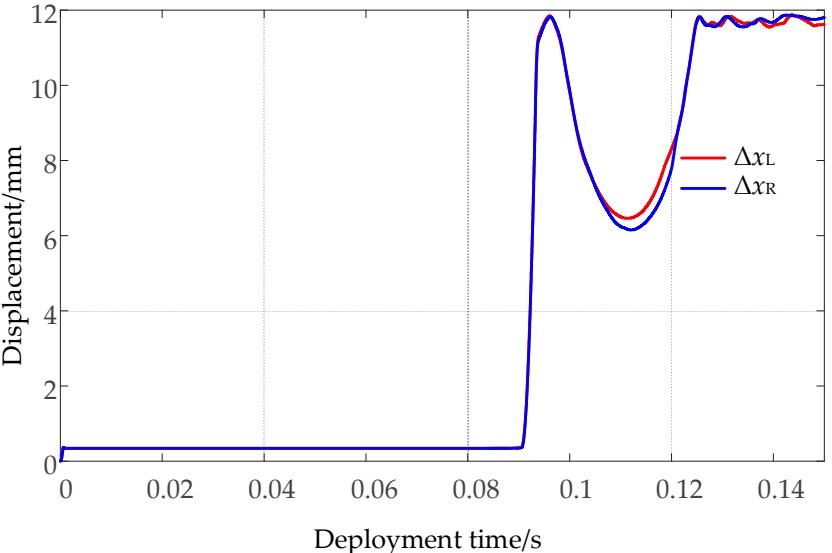

**Figure 8.** The relationships between the displacement of locking springs and deployment time.

The results show that, after triggering the longitudinal locking mechanism, the locking pins are released by the prestressing force of the locking springs and collide with the locking holes. The rebound movement of the locking pin causes the sudden change in its speed. However, under the action of the force of locking spring, the locking pin moves downwards and collides with the inner wing again and again, until the speed drops to zero. The movement of the springs and locking pins on both sides are basically the same. Therefore, it can be concluded that the two pairs of locking mechanisms can be implemented to work synchronously and reliably.

### 3.3. Comparison of Results

Figure 9 compares the theoretical and simulation results. The solid lines represent the theoretical results, while the dotted lines represent the simulation results.

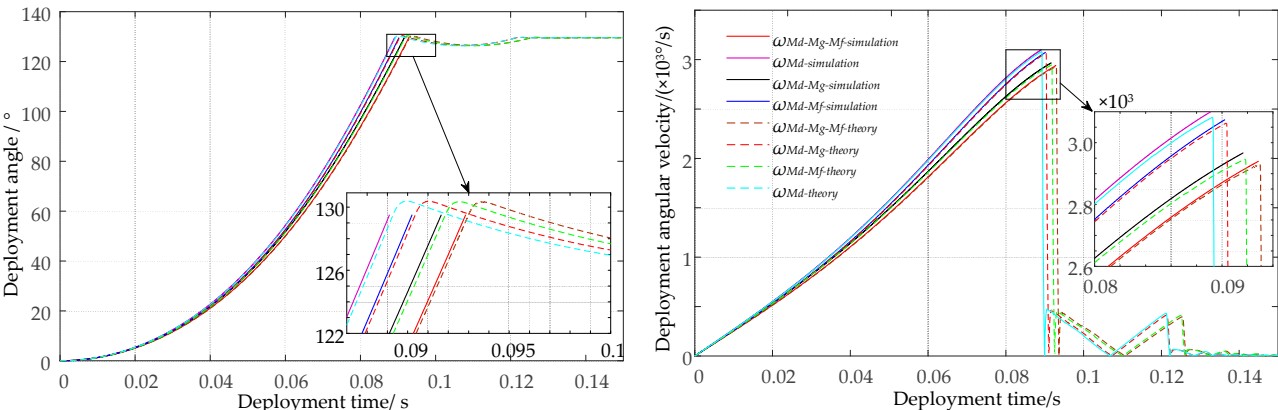

**Figure 9.** Comparison of the theoretical results and simulation results.

The results indicate that the relationships between deployment angle, deployment angular velocity and deployment time in the theoretical and simulation results are similar, with and without consideration of the influence of gravitational torque and frictional torque. This means that the tendency and values of the theoretical results are essentially consistent with the simulation results. Thus, the correctness and accuracy of the dynamic and simulation models of deployable mechanism are both verified.

## 4. Reliability Evaluation

### 4.1. Reliability of Deployable Mechanism

It is well known that the deployment time of folding wing is extremely short (about 100 ms), and any failed action or failed deployable mechanism will cause the folding wing to fail. That is to say, if the deployment time in the specified position is too long, the execution of other actions will be affected significantly. Conversely, if the time is too short, the deployment angular velocity in the specified position of the outer wing will be too high to provide a strong impact on missiles when it collides with the inner wing. Therefore, the reliability of the folding wing requires that each deployable mechanism satisfies the specified performance indexes, such as the specified deployment time $T_F$ and deployment angular velocity $W_F$ in the specified position. Consequently, the reliability of each deployable mechanism can be expressed as

$$R_i = \int_{x_i \subset D} f_i(x_i)dx_i = \int_0^{T_F} \int_0^{W_F} f_i(t_i)f_i(\omega_i)d\omega_i dt_i = R_{t,i} \cdot R_{\omega,i}. \tag{16}$$

where $f_i(t_i)$ and $f_i(\omega_i)$ are the probability density function of the deployment time and deployment angular velocity in the specified position, respectively; $T_F$ is the specified deployment time; $W_F$ is the specified deployment angular velocity.

According to rigid-flexible coupling dynamic analysis, the main factors affecting the deployment time and deployment angular velocity in the specified position are as follows: the position of the pulley $O_2$ ($y_2$, $z_2$) and chute $O_3$ ($y_3$, $z_3$), the equivalent radius $R$ of the pulley and the cable, the friction coefficient $f_v$ between the rotating pair, the prestressing force $f_0$ and the stiffness coefficient $k_s$ of each parallel spring.

As shown in Figure 10, the steps of a new reliability method with angular precision control for rigid-flexible coupling deployable mechanism are shown as follows.

Step 1: Establish the rigid-flexible coupling differential equations.

As shown in Table 2, randomly generate input samples $x_{k,j}$ ($k$ = 1, 2, ..., $m$; j = 1, 2, ..., $N$) based on the parameters of the variables $\{x_1, x_2, ..., x_k, ..., x_m\}^T$. Obtain the differential equations of each deployable mechanism.

**Table 2.** Variables and their parameters.

| Variables | Factors | Distribution Type | Mean | Standard Deviation | Bias | Range |
|-----------|---------|-------------------|------|--------------------|------|-------|
| $x_1$ | $y_2$ | | 6.50 | 0.01 | (−0.03, 0.03) | (6.47, 6.53) |
| $x_2$ | $z_2$ | | −249.50 | 0.01 | (−0.03, 0.03) | (−249.53, −249.47) |
| $x_3$ | $y_3$ | | −26.67 | 0.01 | (−0.03, 0.03) | (−26.7, −26.64) |
| $x_4$ | $z_3$ | | −200.16 | 0.01 | (−0.03, 0.03) | (−200.19, −200.13) |
| $x_5$ | $R$ | Normal distribution | 8 | 0.0167 | (−0.05, 0.05) | (7.95, 8.05) |
| $x_6$ | $f_0$ | | 520 | 10 | (−30, 30) | (490, 550) |
| $x_7$ | $k_s$ | | 5.6 | 0.1 | (−0.3, 0.3) | (5.3, 5.9) |
| $x_8$ | $f_v$ | | 0.1 | 0.0067 | (−0.02, 0.02) | (0.08, 0.12) |
| $x_9$ | $f_r$ | | 4 | 0.0167 | (−0.05, 0.05) | (3.95, 4.05) |

Step 2: Solve the equations with precision control.

Firstly, make $j$ = 1, $i$ = 1. Next, set the range of the numerical simulation time to $[t_0, t_F]$, the initial deployment angle $\theta_0$ and initial deployment angular velocity $\omega_0$ to zero,

and the incremental step depth $\Delta t$ to $10^{-(i+4)}$. Then, ODE 45 is employed to solve the *j*-th equation. Consequently, the relationship between $\theta$, $\omega$ and $t$ is obtained. Lastly, judge the minimum angle $|\Delta\theta|_{\min}$ between $\theta$ and $\theta_p$ to be equal or less than the given angle error $\varepsilon$. If $|\Delta\theta|_{\min} \leq \varepsilon$, output the *j*-th actual angle $\theta_j$ that satisfies the angular precision condition and save the corresponding *j*-th actual deployment time $t_j$ and *j*-th actual deployment angular velocity $\omega_j$. Otherwise, substitute *i* with *i* + 1, and the above step is repeated to solve the equations until the the given angle error $\varepsilon$ is satisfied.

Step 3: Calculate the working reliability of deployable mechanism $R_i$.

Firstly, the distribution types and relevant parameters of actual deployment time and angular velocity in the specified position are confirmed through hypothesis testing. Then, calculate the reliability of time and angular velocity under specified conditions. Finally, multiply $R_{\omega,i}$ and $R_{t,i}$ to obtain the reliability of deployable mechanism.

Step 4: Verify the proposed method by the Monte Carlo method [20].

Firstly, make initial number of failure times $N_F$ to zero, and set the specified deployment time and angular velocity to $T_F$ and $W_F$, respectively. Next, judge whether $t_j$ and $\omega_j$ both fall into the acceptance domain. This means that, if they are both true, $I_F = 0$, otherwise, $I_F = 1$. Then, let $N_F = N_F + I_F$ and judge whether $j = N$ is true or not, which means that the solving of all samples is completed. If they are not equal, set $j = j + 1$ and continue solving. Otherwise, end the loop. Finally, because the result of the Monte Carlo method is often a standard in the theoretical research [21–23], compare the proposed method with the MC method and validate it.

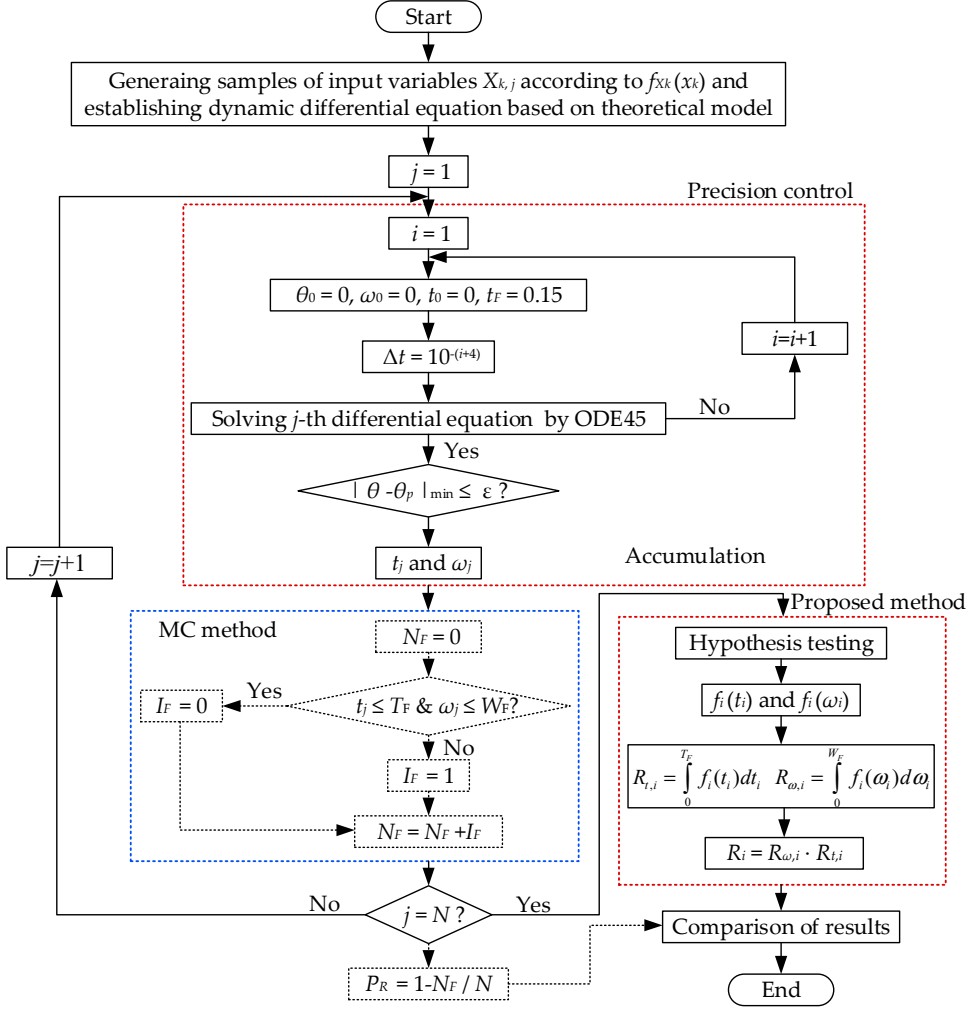

**Figure 10.** Flow chart for reliability method with precision control of deployable mechanism.

$N = 10^5$ samples are carried out to obtain the relevant data. Among them, the red dots represent the lower mechanism, while the blue dots represent the upper mechanism, as shown in Figure 11.

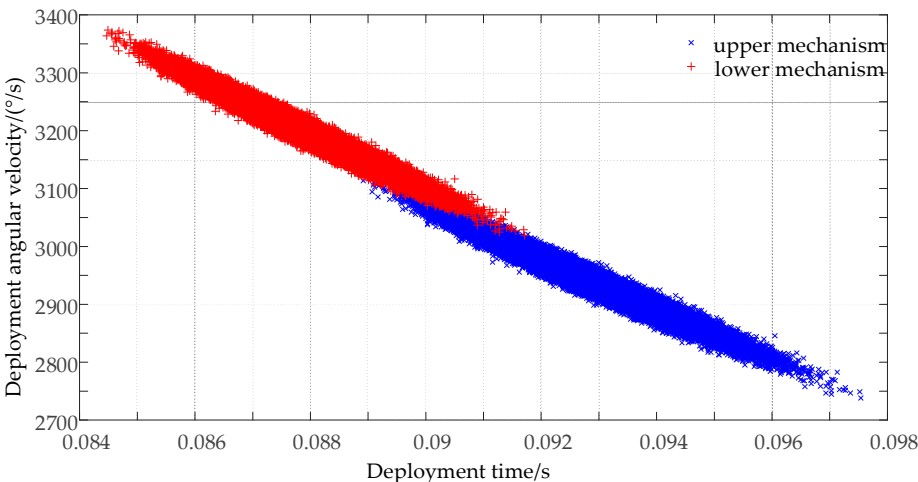

**Figure 11.** Data graph for deployment time and angular velocity in specified position.

The results show that the deployment time in the specified position of the upper mechanism is greater, despite its deployment angular velocity in the specified position being smaller than the lower mechanism. Consequently, both deployment performances need to be considered when calculating reliability.

As shown in Figure 12, a normal distribution with a confidence level of 0.05 is used to fit the discrete dots of the deployment angular velocity of the mechanism, and the kstest method is used to examine the normality. Among them, the red lines represent the lower mechanism while the blue lines represent the upper mechanism.

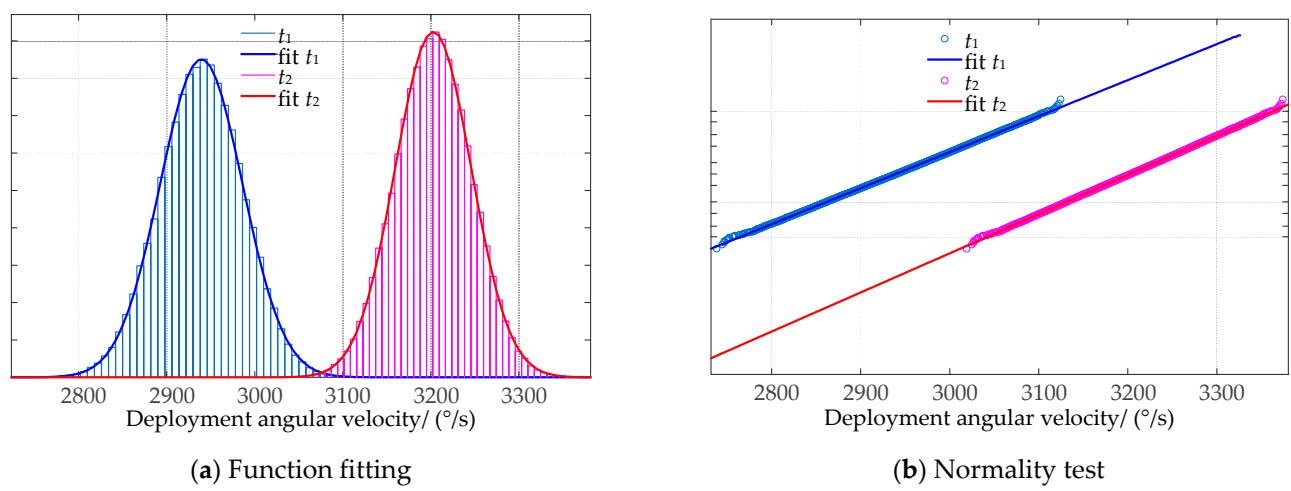

(**a**) Function fitting　　　　　　　　　　　　(**b**) Normality test

**Figure 12.** Function fitting and normality test of deployment angular velocity.

It is obvious that all discrete data follow a normal distribution. The MC method is applied to validate the proposed method under specified conditions, and the reliability of the upper deployable mechanism $R_i$ is shown in Table 3. Furthermore, those of the lower mechanism are similar to the upper mechanism.

**Table 3.** Comparison of results.

| Specified Condition | | Reliability $R_i$ | | Error % |
| --- | --- | --- | --- | --- |
| $T_F$ | $W_F$ | Proposed Method | MC Method | |
| 0.0935 | 2975 | 0.56338 | 0.50484 | 10.391 |
| 0.0935 | 3000 | 0.65453 | 0.63205 | 3.434 |
| 0.094 | 3000 | 0.77471 | 0.76106 | 1.761 |
| 0.094 | 3025 | 0.82970 | 0.82605 | 0.440 |
| 0.0945 | 3025 | 0.90765 | 0.90505 | 0.286 |
| 0.0945 | 3050 | 0.93087 | 0.92898 | 0.203 |
| 0.095 | 3050 | 0.96966 | 0.96842 | 0.127 |
| 0.095 | 3075 | 0.97673 | 0.97519 | 0.158 |
| 0.0955 | 3075 | 0.99191 | 0.99101 | 0.091 |
| 0.096 | 3100 | 0.99823 | 0.99766 | 0.057 |
| 0.0965 | 3125 | 0.99968 | 0.99949 | 0.019 |
| 0.097 | 3150 | 0.99996 | 0.99985 | 0.011 |
| 0.0935 | 2975 | 0.56338 | 0.50484 | 10.391 |

It can be seen that errors of the reliability obtained from the proposed method and the MC method under specified conditions are negligible. That is to say, the assumption that the reliability of time and angular velocity are independent is true, and the proposed method is proved.

*4.2. Reliability of Folding Wing*

It can be recognized that each deployable mechanism is in a series system when the synchronization is not considered. Therefore, the reliability of folding wing is a multiple multiplication of the deployable mechanism $R_i$.

Under normal circumstances, the deployable mechanisms should obey the same distribution of deployment time because they share an identical distribution of geometric parameters, and an identical driving force and physical properties, such as damping and the coefficient of friction. However, different installation positions of the deployable mechanisms and lock mechanisms, and differences in fight attitude would result in significant differences in their working load distribution types or different parameters, which would inevitably result in asynchronous movement.

If the magnitude and the direction of the aerodynamic load, the inertial load acting on each folding wing, different installation positions of the lock mechanisms, the difference in friction coefficient and the driving force because of manufacture errors are excluded, asynchronization occurs between the upper and the lower mechanisms despite their sharing an identical structure, due to the external load factors, such as gravity torque. It is common for several deployable mechanisms to work together to get the missile under control. Asynchronous movement will affect its combat capability or even lead to failure. If the deployment time variance between several sets of deployable mechanisms is too significant, it might cause the missile to become out of control or even lead to a crash. Therefore, synchronization in the reliability of folding wing cannot be ignored.

$R_i$ can be calculated by the reliability of the deployment time and deployment angular velocity, since they are independent according to Section 4.2. Thus, the reliability of folding wing, considering the deployment performance and without considering synchronization, can be expressed as

$$R_s = \prod_{i=1}^{n} R_i = \prod_{i=1}^{n} \left( R_{t,i} \cdot R_{\omega,i} \right) = \prod_{i=1}^{n} R_{t,i} \cdot \prod_{i=1}^{n} R_{\omega,i} = R_t(t \leq T_F) \cdot R_\omega(\omega \leq W_F). \qquad (17)$$

For the synchronization, it should be emphasized that the time reliability, $R_t$, meets the requirements not only for the specified time, but also for synchronization. It is required

that the actual deployment time difference $\Delta t$ is equal to or less than the threshold value $\Delta T$. Therefore, the reliability of folding wing considering synchronization can be rewritten as

$$R_{sys} = R_t(t \leq T_F \& \Delta t \leq \Delta T) \cdot R_\omega(\omega \leq W_F). \tag{18}$$

That is to say, when the specified time of the upper deployable mechanism is determined, the time of the lower deployable mechanism should be within the range of $(\max(t_1 - \Delta T, 0), \min(t_1 + \Delta T, T_F))$. Consequently, the time reliability can be expressed as

$$R_t = \int_0^{T_F} f_1(t_1) \left[ \int_{\max(t_1 - \Delta T, 0)}^{\min(t_1 + \Delta T, T_F)} f_2(t_2) dt_2 \right] dt_1. \tag{19}$$

If $\Delta T < t_1$, $\max(t_1 - \Delta T, 0) = t_1 - \Delta T$; otherwise, $\max(t_1 - \Delta T, 0) = 0$. Similarly, if $t_1 < \Delta T + T_F$, $\min(t_1 + \Delta T, T_F) = t_1 + \Delta T$; otherwise, $\min(t_1 + \Delta T, T_F) = T_F$.

As shown in Figure 13, $f_1(t_1)$ and $f_2(t_2)$ are the probability density functions of deployment time of the upper and lower mechanisms, respectively.

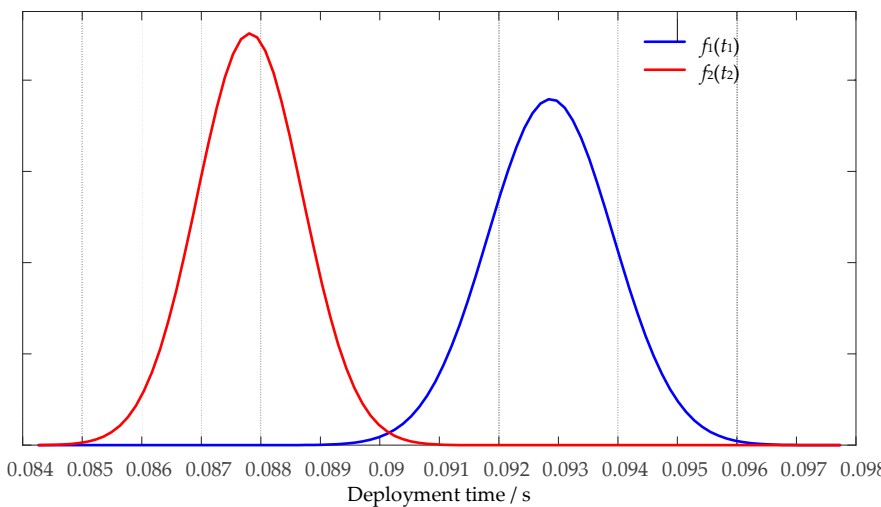

**Figure 13.** The probability density functions of deployment time of deployable mechanisms.

In order to compare the effects of deployment performance or synchronization on time reliability of folding wing, time reliability is calculated under different conditions, as shown in Table 4. $R_{t,1}$, $R_{t,2}$, $R_{t,3}$ or $R_{t,4}$ represent the situations where only the specified time $T_F$, only the time difference $\Delta T$, and both the specified time and the time difference are considered. The difference between $R_{t,3}$ and $R_{t,4}$ is the calculation method, $R_{t,3}$ is based on Formula (19), while $R_{t,4}$ is the multiplication of $R_{t,1}$ and $R_{t,2}$.

**Table 4.** Comparison of time reliability.

| Deployment Performance $T_F$/ms | Synchronization $\Delta T$/ms | Methodology | | | | Error % |
|---|---|---|---|---|---|---|
| | | $R_{t,1}$ | $R_{t,2}$ | $R_{t,3}$ | $R_{t,4}$ | |
| 95 | 5.5 | 0.97685 | 0.99440 | 0.97685 | 0.97138 | 0.56 |
| 95 | 5.4 | 0.97685 | 0.98077 | 0.97623 | 0.95807 | 1.86 |
| 94 | 5.5 | 0.85841 | 0.99440 | 0.85841 | 0.85360 | 0.56 |
| 94 | 5.2 | 0.85841 | 0.80969 | 0.81198 | 0.69505 | 14.40 |
| 94 | 5.0 | 0.85841 | 0.38450 | 0.38446 | 0.33006 | 14.15 |
| 93 | 5.1 | 0.55622 | 0.61034 | 0.55622 | 0.33948 | 38.97 |
| 92 | 5.0 | 0.20687 | 0.38450 | 0.20687 | 0.07954 | 61.55 |
| 92 | 4.9 | 0.20687 | 0.18065 | 0.18033 | 0.03737 | 79.28 |
| 95 | 5.5 | 0.97685 | 0.99440 | 0.97685 | 0.97138 | 0.56 |

The results show that errors of time reliability between the values obtained from Formula (19) and the multiplication are quite distinct. That is to say, both deployment performance and synchronization cannot be ignored when calculating the reliability of the folding wing. Moreover, time reliability is not the multiplication of $R_{t,1}$ and $R_{t,2}$.

## 5. Discussion

In view of the lack of studies and time-consuming calculation of the reliability of the rigid-flexible coupling folding wing, we have attempted to establish the theoretical and simulation dynamic model of the deployable mechanism, and to verify its accuracy and precision. Against this background, a new reliability method with angular precision control for the cable-spring deployable mechanism is proposed and verified by the MC method. Therefore, the reliability of the folding wing, considering deployment performance and synchronization, is analyzed.

However, the proposed reliability evaluation of the folding wing is not sufficiently exhaustive to cover the synchronization in the whole deployment stage. In addition, the proposed reliability evaluation is not universal enough to simultaneously cover the asynchronization of the left and right wings caused by external loads, such as aerodynamic torque. The reliability of folding wing resulting from these problems will be considered in future work.

**Author Contributions:** Conceptualization, Y.G. and M.H.; methodology, Y.G.; software, Y.G.; validation, Y.G. and M.H.; formal analysis, M.H.; investigation, Y.G.; writing—original draft preparation, Y.G.; writing—review and editing, Y.G. and M.H.; supervision, M.H and X.Z.; project administration, Y.G.; funding acquisition, M.Z. and X.Z. All authors have read and agreed to the published version of the manuscript.

**Funding:** This research was funded by the Program for the National Key Research and Development Plan Project (Grant No. 2018YFB1308100), National Natural Science Foundation of China (Grant No. 51375458).

**Institutional Review Board Statement:** Not applicable.

**Informed Consent Statement:** Not applicable.

**Data Availability Statement:** The raw/processed data required to reproduce these findings cannot be shared at this time as the data also forms part of an ongoing study.

**Acknowledgments:** Thanks Jing Yang and Xingwen Gao for fruitful discussion and assistance.

**Conflicts of Interest:** The authors declare no conflict of interest.

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
