# Peer review of "Reliability Evaluation for Cable-Spring Folding Wing Considering Synchronization of Deployable Mechanism"

_actuators, doi:10.3390/act10050099_

Round 1
Reviewer 1 Report
In this paper the authors applied a methodology that allows studying the reliability of the developed cable-spring Folding Wing. The paper is very interesting and is based on a sound approach. I can recommend it in its present form. The multibody model developed in Adams is well detailed as well as the comparison of theoretical and simulation results. The reliability estimated by the proposed method appears solid and fits very well the MC method results. Maybe, in order to help the reader better understand the working principle (see par. 2.1), more details on the deployment mechanism could be given in par. 2.1 by including CAD images at different deployment time.
Reviewer 2 Report
The authors present a conceptual design of a folding wing mechanism for missiles and rockets as well as a new evaluation methodology. It is a nice engineering paper, but it doesn't quite reach to the level of scientific research paper.
It is bad practice to introduce two new concepts in one paper. The new evaluation methodology should be verified on a known reference geometry, and a new concept should be tested with known methods (this is arguably done in the paper with the MC method)
Validation of the realiability evaluation should be done on experimental data, not numerical.
Correlating deployment time variance directly to reliability needs to be motivated much more strongly as well as clearly defining what other unreliability sources are excluded. (e.g. variance in aerodynamic and inertial loads. )
The language can be improved, there are some phrasing that needs looking over. E.g. the split infinitives on row 30.
Reviewer 3 Report
This paper presented modeling and reliability evaluation of cable-spring folding wing considering synchronization of deployable mechanism. The paper is well structured and the modeling and simulation results were clearly explained. The followings are some comments for the authors to improve the manuscript.
1) Since it will be a journal publication, experimental validation is usually required. The current manuscript only contain simulation results. It seems insufficient.
2) A figure showing the basic structure of the proposed mechanism will help the readers to better understand the working principle of the mechanism. A sketch will be fine. Figure 1 in the current manuscript does not clearly explain the cable-spring folding wing mechanism.
3) Figure 3 showed the equivalent model of the deployable mechanism. How is it equivalent with the mechanism? Without a clear explanation of the original mechanism, it is hard for the readers to understand this equivalence.
4) English presentation could be improved significantly. Academic writing style should be followed.
5) The labels in Figure 5 are too small to read. Usually, figures in academic paper should be self-explained. More efforts should be put in preparing the figures.
6) In abstract, line13, the size ---> the magnitude? We do not usually use size to describe the magnitude of force.
7) In abstract, line 23, contrasting with ---> comparing with
8) Line 102, that in Reference 2 ---> that in [2]
9) Line 138, In the formula, ---> where
10) Line 319, need to considered in ---> need to be considered
11) Line 328, the prosed method under different specified condition ---> the proposed method under different specified conditions
Round 2
Reviewer 2 Report
The authors clearfied any concerns with the revision.
Author Response
Thank for your positive comments and suggestions.
Reviewer 3 Report
The paper has been improved after revision. Some minor comments on the presentation style could be further improved as below.
1. For literature citation, if one wants to mention the authors' name of the referred paper, the first author's family name is usually used instead of full name. For example, Kroyer designed a novel.... If the literature includes more than one author, it should be Kroyer et al. designed a novel....
2. When writing equations, the equation should be considered as a part of a sentence. For example, equation (3) could be presented as
..., the driving force Fs can be expressed as
Fs = m(), (3)
where n is the number...
Note, a comma follows the equation and 'where' is a subordinate clause to explain the variables included in the equation. Therefore, it should be low-cased without indentation.
I suggest the authors to use professional proofreading to correct the English presentation.
